# CAR-T Cells with Phytohemagglutinin (PHA) Provide Anti-Cancer Capacity with Better Proliferation, Rejuvenated Effector Memory, and Reduced Exhausted T Cell Frequencies

**DOI:** 10.3390/vaccines11020313

**Published:** 2023-01-31

**Authors:** Gamze Gulden, Berranur Sert, Tarik Teymur, Yasin Ay, Nulifer Neslihan Tiryaki, Abhinava K. Mishra, Ercument Ovali, Nevzat Tarhan, Cihan Tastan

**Affiliations:** 1Molecular Biology, Institute of Science and Technology, Üsküdar University, Istanbul 34662, Turkey; 2Transgenic Cell Technologies and Epigenetic Application and Research Center (TRGENMER), Üsküdar University, Istanbul 34662, Turkey; 3Molecular Biology and Genetics Department, Faculty of Engineering and Natural Science, Üsküdar University, Istanbul 34662, Turkey; 4Molecular, Cellular and Developmental Biology Department, University of California Santa Barbara, Santa Barbara, CA 93106, USA; 5Acıbadem Labcell Cellular Therapy Laboratory, Istanbul 34752, Turkey; 6Faculty of Humanities and Social Sciences, Üsküdar University, Istanbul 34662, Turkey

**Keywords:** immunotherapy, leukemia, CAR-T, immunologic memory, phytohemagglutinin

## Abstract

The development of genetic modification techniques has led to a new era in cancer treatments that have been limited to conventional treatments such as chemotherapy. intensive efforts are being performed to develop cancer-targeted therapies to avoid the elimination of non-cancerous cells. One of the most promising approaches is genetically modified CAR-T cell therapy. The high central memory T cell (Tcm) and stem cell-like memory T cell (Tscm) ratios in the CAR-T cell population increase the effectiveness of immunotherapy. Therefore, it is important to increase the populations of CAR-expressing Tcm and Tscm cells to ensure that CAR-T cells remain long-term and have cytotoxic (anti-tumor) efficacy. In this study, we aimed to improve CAR-T cell therapy’s time-dependent efficacy and stability, increasing the survival time and reducing the probability of cancer cell growth. To increase the sub-population of Tcm and Tscm in CAR-T cells, we investigated the production of a long-term stable and efficient cytotoxic CAR-T cell by modifications in the cell activation-dependent production using Phytohemagglutinin (PHA). PHA, a lectin that binds to the membranes of T cells and increases metabolic activity and cell division, is studied to increase the Tcm and Tscm population. Although it is known that PHA significantly increases Tcm cells, B-lymphocyte antigen CD19-specific CAR-T cell expansion, its anti-cancer and memory capacity has not yet been tested compared with aCD3/aCD28-amplified CAR-T cells. Two different types of CARs (aCD19 scFv CD8-(CD28 or 4-1BB)-CD3z-EGFRt)-expressing T cells were generated and their immunogenic phenotype, exhausted phenotype, Tcm–Tscm populations, and cytotoxic activities were determined in this study. The proportion of T cell memory phenotype in the CAR-T cell populations generated by PHA was observed to be higher than that of aCD3/aCD28-amplified CAR-T cells with similar and higher proliferation capacity. Here, we show that PHA provides long-term and efficient CAR-T cell production, suggesting a potential alternative to aCD3/aCD28-amplified CAR-T cells.

## 1. Introduction

The development of genetic modification techniques has opened a new era in cancer treatments that were limited to conventional treatments such as chemotherapy and monoclonal antibodies. The main reason is that the host immune cells cannot be sufficiently activated by cancer cells to exert a cytotoxic function due to low anti-tumor activity or lack of effector T cells [1]. The most promising of these methods is genetically modified CAR-T cell therapy. The first clinical studies using transgenic CAR-T cells were in patients with hematological cancers, including B-cell acute lymphoblastic leukemia (ALL), Non-Hodgkin’s Lymphoma (NHL), chronic lymphocytic leukemia (CLL), and Multiple Myeloma (MM). This demonstrated improved response rates ranging from 50% to 85%, significant disease-free, and overall survival [2,3]. The US Food and Drug Administration (FDA), European Union, and Canada have approved Kymriah (Tisagenlecleucel) by Novartis for pediatric and young adult patients with ALL; also approved is Yescarta (Axicabtagene ciloleucel) by Kyte Gilead for the treatment of adults with relapsed or refractory large B-cell lymphoma. 

Chimeric antigen receptors are engineered receptors that typically contain the antigen binding site of a monoclonal antibody (mAb), T cell receptor transmembrane domain, and an intracellular signaling domain of the CD3ζ chain. Following initially disappointing results with the first generation of CAR-T cells, the cytotoxicity, expansion, and persistence of CAR-T cells in clinical studies have been improved in subsequent generations of CARs by including one or more intracellular domains of co-stimulatory molecules, such as CD28 or 4-1BB [4,5,6]. Trials with CARs containing CD28 or 4-1BB domains have shown similar initial response rates in patients with ALL [3,7,8]. Long-term expression and effector function of CAR (CD28)-T cells were observed in in vivo studies at 30 days. In contrast, the prolonged expressions of the effector function of CAR (4-1BB)-T cells have been reported to persist for up to 4 years in patients [9]. In addition, the inclusion of 4-1BB signaling domains in some CARs has been shown to attenuate immunodeficiency (exhaustion) in CAR-T cells [10]. Endogenous 4-1BB signaling has been reported to have long-term survival benefits for T cells [11]. 

The prolonged proliferation and durability that central memory T cells (Tcm) and stem cell-like memory T cells (Tscm) induce following T cell therapy indicate crucial preconditions for the effectiveness of the treatment [12]. Studies have shown that the high proportions of central memory T cells and stem cell-like memory T cells in the CAR-T cell population provide the necessary prerequisites (continuous proliferation and long-term persistence) for the effectiveness of immunotherapy [12,13]. Tcm and Tscm cell populations were found to be higher in the 4-1BB CAR group compared with the CD28 CAR group. In contrast, an increase in the T cell group with a high effector memory phenotype (Effector memory T cell, Tem) was observed in CD28 CAR cultures [11,14]. A crucial phase of the adaptive immune response is T cell activation. In addition to producing cytotoxic T cell responses, activation is necessary for efficient CD4 T cell responses [15]. Activated T cells may proliferate, differentiate, secrete cytokines, kill target cells, and perform other effector tasks [16]. Clinically applicable conventional T cell stimulation in vitro requires multivalent anti-CD3 and anti-CD28 antibodies [17].

As a good alternative, Phytohemagglutinin (PHA) binds to the TCR/CD3 complex, mimicking all intracellular activation events triggered by anti-CD3 antibodies [18]. PHA is a lectin derived from red kidney beans that bind to the membranes of T cells and stimulates metabolic activity and cell division [19]. PHA is a common polyclonal stimulant used in retroviral transduction protocols [20]. Human CD8+ T cells are known to be difficult to culture for extended periods (>3 weeks) in culture. The optimal viability of CD8+ T cells can be maintained for up to 21 days with IL-15 in culture, but studies have shown that CD8+ T lymphocytes can be cultured for a long time (up to 90 days) by combining IL-15 and PHA [21]. PHA induced a marked increase in the CD8+ Tcm population and CD8+ TN cells, more importantly the CD8+ CCR7 Tem and TEf subsets exhibited the effector phenotype including cytotoxic capacity and performance expression [22]. Despite inducing cell proliferation, PHA also promotes cell apoptosis due to FasL interactions [23], which can be inhibited by IL-15 [24]. Thus, the combination of PHA and IL-15 signals creates a favorable environment for the growth and expansion of CD8+ T cells.

In this research, Tcm and Tscm population ratios and cytotoxic activity were assessed in the PHA- or aCD3/aCD28-activated CAR-T cells with two different CAR (aCD19 scFv-CD8-(CD28 and/or 4-1BB)-CD3z-EGFRt) constructs. The efficiency was determined in CAR-T cell production and proliferation by adding IL-2, IL-7, IL-15 cytokines, and PHA in the medium. It is shown that cultured CAR-T cells with PHA provide Tcm and Tscm profiles to these cells so that they can remain long-term and cytotoxic in in vitro cancer models. With this research, the long-term and effective CAR-T cell production method can be tested in animal experiments. 

## 2. Materials and Methods

### 2.1. Synthesis of CAR Construct and Lentivirus Production

The lentiviral vector encoding the CD19-CD28z-specific CAR (CD19-CD8-CD28-CD3z) was designed and synthesized by Creative Biolabs (New York, NY, USA). The lentiviral vector encoding CD19-specific CAR (CD19-CD8-4-1BB-CD3z) was designed and synthesized by GenScript. The envelope pCMV-VSV-G plasmid (from Bob Weinberg (Addgene #8454; http://n2t.net/addgene:8454 accessed on 9 April 2003; RRID: Addgene_8454)) and psPAX2 plasmid (from Didier Trono (Addgene #12260); http://n2t.net/addgene:12260, accessed on 15 December 2020; RRID: Addgene_12260)) required for lentivirus production were obtained from Addgene. Plasmids were amplified by integrating CAR-encoding plasmids to be used for genetic modification and plasmids (VSVG and psPAX2) used to produce lentivirus packaging proteins into *E. coli* DH5α (NEB C2987H) strain. Plasmid DNA was produced by using CompactPrep Plasmid Maxi Kit (QIAGEN, Cat No: 12863) and Zymopure Plasmid Maxiprep Kit (Zymopure, Cat: D4202). DNA concentration was measured in ng/µL with Microplate ELISA Reader (FLUOstar Omega) and it was evaluated whether the purity value was between 1.8 < A260/A280 < 2.0. For the control of the isolated plasmids, DNA samples loaded on the 1% agarose gel prepared in 1X TAE buffer solution using a BIO-RAD gel electrophoresis system were run at 90 V and 60 min. Plasmid DNA samples with <1% of bacterial DNA contamination were used for lentivirus production. Isolated envelope, packaging, and CAR plasmids were treated with FuGENE (Promega, Madison, WI, USA, E2311) transfection reagent and then lentivirus production was performed using host cells HEK293T (10% FBS and 1% penicillin/streptomycin, Gibco DMEM HG medium with L-Glutamine). Packaged recombinant lentiviruses were collected 72 h after transfection from the supernatant of HEK293T cell cultures. Produced CAR lentiviruses were concentrated with the Lenti-X Concentrator (Takara Bio, Shiga, Japan, 631232) to increase virus concentration (20X–100X). Viruses were stored at −80 °C [25].

### 2.2. Lentivirus Titration

The Jurkat cell line was suspended as 10,000 cells per well in 100 µL of RPMI with glutamine HEPES including 10% FBS, 1% pen/strep, 1% non-essential amino acids, 1% sodium pyruvate, and 1% vitamins. Jurkat cells in 100 µL of medium were plated in U bottom 96-well plates from A to H. The wells were adjusted to have 10 µL, 3 µL, 1 µL, 0.3 µL, 0.1 µL, 0.03 µL, and 0.01 µL of the 100x-concentrated CAR-LV solutions in each 50 µL of the medium, and then 50 µL of virus dilution from each concentration was transferred to Jurkat cultured wells, the total volume was adjusted to 150 µL, and cells were incubated for 72 h. EGFRt expression was determined using Cytoflex Flow Cytometer (Beckman Coulter, Brea, CA, USA, B5-R3-V0) with an anti-EGFR-FITC antibody (R&D Systems, Minneapolis, MN, USA, FAB10951G).

### 2.3. T Cell Transduction and CAR-T Culture Conditions

The research was approved by the Acıbadem University and Acıbadem Health Institutions Medical Research Ethics Committee (ATADEK-2019-17/31). Healthy adult blood samples were obtained and peripheral blood mononuclear cell (PBMC) isolation was performed at Uskudar University, Transgenic Cell Technologies and Epigenetics Application and Research Center (TRGENMER). Blood samples from three healthy human donors were combined with Ficoll (Paque PREMIUM 1.073, 17-5446-52). PBMCs were isolated by the density-gradient centrifugation method. For the isolation of CD4+ and CD8+ T cells from PBMC, CD3+ T cell isolation was performed by using anti-CD3 microspheres (Miltenyi Biotech, Bergisch Gladbach, Germany, 130-050-101) instead of anti-CD4 and anti-CD8 microspheres. Total CD3+ T cells were isolated using anti-CD3 microspheres (Miltenyi Biotech, 130-050-101). Initial T cell activation was performed with anti-CD3/anti-CD28 microbeads (T Cell TransAct, Miltenyi Biotech, 130-111-160) and 10µg/mL Phytohemagglutinin-M (PHA-M) (Roche 11082132001, 20 mg). Lentivirus transduction procedure, Vectofusin 1 (10 µg/mL) (Miltenyi Biotech, 130-111-163), and BX-795 hydrochloride (6 µM) (Millipore Sigma, Burlington, MA, USA, SML0694- 5MG) were inoculated with 2-3 MOI lentiviruses. Cells were cultured in T cell medium (50 IU/mL IL-2, 10 ng/mL IL-7, 5 ng/mL IL-15, 10% Fetal Bovine Serum, and 1% pen/strep) for 14 days. CAR expression levels were determined by Cytoflex Flow Cytometer analysis using an anti-EGFR-FITC antibody. The second activation and expansion (re-stimulation) study on day 14 was performed using PHA and aCD3/aCD28. Re-transduction assay was performed using Vectofusin-1 with 5–10 MOI lentiviruses 24 h after the second activation [25].

### 2.4. Analysis of T Cell Sub-Populations (Tn-Tcm-Tem-Tef) and Exhaustion Profile

T cell sub-population profiling studies on days 14 and 21 were analyzed using CD3-PC7 (Beckman Coulter, 6607100), CD4-APC-A700 (Beckman Coulter, B10824), CD8-PC5.5 (Beckman Coulter, B21205), EGFR-A488 (R&D Systems, FAB10951G), CD45RA-ECD (Beckman Coulter, IM2711U), CD45RO-PE (Miltenyi Biotech,130-113-559), CD27-APC-A750 (Beckman Coulter, B12701), and CD62L-APC (Miltenyi Biotech, 130-113-617) antibodies by Cytoflex Flow Cytometer analysis. Exhaustion profiling studies on days 14 and 21 were analyzed using CD3-PC7 (Beckman Coulter, 6607100), CD4-APC-A700 (Beckman Coulter, B10824), CD8-PC5.5 (Beckman Coulter, B21205), EGFR-A488 (R&D Systems, FAB10951G), CD279 (PD1)-PE (Miltenyi Biotech, 130-117-384), CD366 (TIM3)—APC (Invitrogen, Waltham, MA, USA, 17-3109-42), and CD223 (LAG-3)—APC-eFluor 780 (Invitrogen, 47-2239-42) antibodies by Cytoflex Flow Cytometer analysis.

### 2.5. In Vitro Anti-Tumor Cytotoxicity and Efficacy Assay

In vitro studies with CAR-T cells (CD19-CD28z and CD19-BBz) were performed for the assessment of efficacy and cytotoxic capacity. For that purpose, anti-CD19-expressing CAR-T cells and CD19-expressing RAJI cells were cultured for 24 h, 7 days, and 14 days (CAR-T: RAJI; 1:1, 5:1, 10:1). Anti-cancer profiling studies on days 14 and 21 were analyzed using CD3-PC7 (Beckman Coulter, 6607100), CD4-APC-A700 (Beckman Coulter, B10824), CD8-PC5.5 (Beckman Coulter, B21205), EGFR-A488 (R&D Systems, FAB10951G), CD19-ECD (Beckman Coulter, A07770), CD25-APC (Miltenyi Biotech, 130-113-284), and CD107a (LAMP-1) -PE (Miltenyi Biotech, 130-111-621) by Cytoflex Flow Cytometer analysis. In the co-culture experiments, the death of CD19+ RAJI cells after 48 h and the CD25 activation (IL2RA, IL-2 receptor alpha chain) and CD107a (marker for degranulation of lymphocytes) cytotoxic de-granulation biomarkers of CAR-T cells and control T cells in CD3+ T cells were analyzed by flow cytometry. Survival analysis of CAR-T cells was performed to control the cell viability of RAJI cells and CAR-T cells. Trypan blue (Biological Industries, #03-102-1B) was applied to identify and count surviving cells. Cell counting and viability analysis were performed with the BIO-RAD TC20 Automated Cell Counter.

### 2.6. Statistics Analysis

Two-tailed Homoscedastic *t*-tests were performed using SPSS software. Outliers were not excluded in any of the statistical tests and each data point represents an independent measurement. Bar plots report the mean and standard deviation or the standard deviation of the mean. The threshold of significance for all tests was set at * *p* < 0.05. *ns*: non-significant.

## 3. Results 

### 3.1. In Vitro Assessment of CAR-T Cell Proliferation, Differentiation, and Anti-Cancer Capacity

The activation of T cells was performed by PHA (10 µg/mL) or by anti-CD3/anti-CD28 T cell Transact™ human (Miltenyi Biotech, 130-111-160). We asked whether PHA increases the Tcm and Tscm population compared with the anti-CD3/anti-CD28 microbeads. CAR-T cell production, activity, and sub-population profiling following T cell activation with PHA were determined (Figure 1A). Based on the Tcm–Tscm ratios we observed, we aimed to develop the CAR-T cell production protocol with CD3 isolation. We successfully achieved >99% CD3+ T cell isolations with three different donors (Figure 1B). Next, we tested the lentiviral transduction capability of PHA-induced T cells with CAR1928 or CAR19BB expression. CAR expressions of CAR-T cells obtained by culturing CD3+ T cells transduced with CAR19BB and CAR1928 lentiviruses with Anti-EGFR-A488 antibody were analyzed by flow cytometry. CAR-T cells activated with anti-CD3/anti-CD28 and PHA and transduced with CAR19BB had an expression frequency of 22.31 ± 1.5% and 22.06 ± 5.9%, respectively. CAR-T cells activated with anti-CD3/anti-CD28 and PHA and transduced with CAR1928 had an expression frequency of 19.53 ± 2.0% and 12.70 ± 0.23%, respectively (Figure 1C). There was not a significant difference (two-tailed *t*-test). Although we approached ~20% CAR expression with anti-CD3/anti-CD28-activated CAR1928 T cells, we could not achieve this with PHA (±20% standard deviation). We showed that PHA-induced T cells can be transduced with CAR-encoding lentiviruses without statistical difference regarding CAR expression on the aCD3/aCD28-induced T cells. Afterward, we asked whether we can increase CAR expression using 2nd lentiviral transduction after reactivation of the T cells with the same activation reagents including aCD3/aCD28 and PHA. MOI ratios were increased from 2 to 3 MOIs (primary transduction) to 5–10 MOIs to provide more than 20% CAR expression in cells in a second transduction process. The second activation and expansion (re-stimulation) study on day 14 was performed using PHA and aCD3/aCD28. PHA/aCD3/aCD28-activated CD3+ T cells were reactivated with PHA/aCD3/aCD28. The second transduction was performed 24 h after the reactivation. Thus, we tested whether the use of 5–10 MOI of CAR lentivirus would increase our expression by second transduction. When the CAR expressions in T cells were assessed, it was observed that the PHA-activated T cells expressed 84.8 ± 12.3% CAR while anti-CD3/anti-C28-activated T cells expressed 62.6 ± 4.1% without a statistical significance (Figure 1D). We showed that both T cell restimulation approaches can achieve upregulation of the CAR expression with 2nd transduction. This suggests that there is no difference in the production of CAR-T cells with lentiviral transduction after the second stimulation. Thus, when re-activated with PHA, CAR-T cell expression increased significantly compared with anti-CD3/anti-C28-activated T cells. Next, we investigated the proliferation capacity of CAR-T cells produced with PHA compared with the anti-CD3/anti-CD28-activated CAR-T cells. The graph of the initial CAR-T cell count, activated with PHA and anti-CD3/anti-CD28, on day 0 and the division coefficient of cell proliferation on day 14 was determined. Here we show that the CAR-T cell proliferation capacity is significantly higher when the cells are stimulated with PHA (* *p* = 0.0098, Figure 1E).

### 3.2. Immunoprofiling Sub-Populations of The Activated T Cell (Tn-Tcm-Tem-Tef)

The activated T cells differentiate to develop subtypes of effector and memory T cells. Next, we aimed to determine T cell sub-population frequencies upon activation with different stimulation reagents including PHA and aCD3/aCD28. To immune profile the activated T cells, we determined the Tcm–Tscm sub-populations using conventional immune biomarkers including Tn + Tscm (CD3^+^, CD45RA^+^, CD62L^+^), Tcm (CD3^+^, CD45RA^−^, CD62L^+^), TemEarly (CD3^+^, CD45RO^+^, CD27^+^, CD45RA^−^, CD62L^−^) T cells, TemLate (CD3^+^, CD45RO^+^, CD27^−^, CD45RA^−^, CD62L^−^) and Temra (CD3^+^, CD45RA^+^, CD62L^−^) [26,27,28,29] (Figure 2).

In the Tn + Tscm, Tcm, TemEARLY, TemLATE, and Temra sub-populations of total CD4+ T cells among PHA-activated CD3+ T cells, the Tn + Tscm ratio increases until day 14 (*p* = 0.036, Figure 3A). On day 14, the Tn + Tscm ratio increased, however, on day 21 after the second activation, the Tn + Tscm ratio decreased significantly (*p* = 0.002, Figure 3A). While the Tcm rate decreased until day 14, it increased statistically on day 21 after the second activation. Compared with day 7 and day 14, the rate of TemEARLY increases especially in the reactivation (*p* = 0.0005 and *p* = 0.0002, Figure 3A). While the Temra rate increases slightly until day 14 (*p* = 0.02), it decreases on day 21 (*ns*, Figure 3A). In CD8+ T cells among PHA-activated CD3+ T cells, the Tn + Tscm ratio increased until day 14 after the first activation (*p* = 0.022); while the Tcm rate decreased on day 14 compared with day 7 (*p* = 0.0003, Figure 3B). TemEARLY increased after the reactivation (*p* = 0.0005 and *p* = 0.0003, Figure 3B). TemLATE also increased on day 21 compared with day 7 (*p* = 0.022, Figure 3B). An increase was observed in Temra until day 14 (*p* = 0.002, Figure 3B). Statistically, there was also an increase in the rate of Temra in reactivation (*p* = 0.03, Figure 3B). Therefore, the increase in the exhausted PHA-activated total CD3+ T cells seems to have occurred after the reactivation. To assess the frequency of the sub-populations of total CD8+ T cells in CD3+ T cells activated with anti-CD3/anti-CD28, the Tn + Tscm ratio increased on day 14 compared with day 7 (*p* = 0.002, Figure 3C). Then, on day 21, this rate decreased (*p* = 0.00001, Figure 3C). The Tcm rate decreased on day 14 and day 21 days compared with day 7 (Figure 3C). TemLATE and Temra rates increased statistically on day 14 and day 21 compared with day 7 (*p* = 0.0005 and *p* = 0.0009, Figure 3C). To determine the populations of total CD4+ T cells in anti-CD3/anti-CD28-activated CD3+ T cells, the Tcm rate decreased on day 14 compared with day 7 (*p* = 0.0003, Figure 3D). The Tcm rate increased on day 21 compared with day 14 (*p* = 0.003, Figure 3D). Although the Tn + Tscm ratio increased on day 14 after the first activation compared with day 7 (*p* = 0.00008), it decreased on day 21 after the reactivation (*p* = 0.00002, Figure 3D). While the rate of TemLATE increased from day 7 to day 14 (*p* = 0.04), it decreased on day 21 after the reactivation (*p* = 0.00006, Figure 3D). The Temra rate also increased on day 14 and day 21 compared with day 7 (*p* = 0.006 and *p* = 0.03, Figure 3D).

Next, we compared the PHA and anti-CD3/anti-CD28-activated total CD4+ T cell and CD8+ T cell in CD3+ T cell populations isolated from three healthy donors. There was no statistical difference for Tn + Tscm, Tcm, TemEARLY, and Temra except TemLATE in total CD4+ T cells (Figure 4A). The only difference was determined in the frequency of TemLATE with a decrease upon activation with PHA (*p* = 0.002, Figure 4A). When the effect of PHA and anti-CD3/anti-CD28 activation are compared, the Tn + Tscm ratio was significantly increased while the TemLATE frequency was impaired in the total PHA-induced CD8+ T cells (*p* = 0.02, Figure 4B). These data suggest that PHA-activated cells differentiate more into early memory compared with that with anti-CD3/anti-CD28 activation. Next, we aimed to investigate the change in the distribution of CD4+ and CD8+ T cell sub-types reactivated with PHA and anti-CD3/anti-CD28 on day 21. We determined that the Tn + Tscm population significantly decreased when the cells were activated with PHA (*p* = 0.002, Figure 4C). Although there was no change in the Tcm population, PHA-activated TemEARLY and TemLATE populations were higher (Figure 4C). When we determined the level of exhaustion markers on Temra, PHA-induced cells had no statistical difference regarding the anti-CD3/anti-CD28 cells (Figure 4C). On the other hand, the ratio of Tn + Tscm and Tcm in total PHA-activated CD8+ T cells was also not significantly different from the anti-CD3/anti-CD28-activated cells (Figure 4D). However, the level of TemEARLY increased in the PHA-activated population (*p* = 0.005, Figure 4D). Moreover, the rate of TemLATE was significantly high in the anti-CD3/anti-CD28-activated population (*p* = 0.015, Figure 4D). These data suggest that PHA supports early memory phenotypes in the reactivated T cells.

Next, we wanted to determine the effect of CAR1928 or CAR19BB expression on T cell differentiation and subtypes. We determined subtype frequencies of the CD4+ or CD8+ T cells expressing CAR constructs including CD28 or 4-1BB costimulatory domain. On day 7 and day 14, there was no significant change in the subtype populations of the CAR-T cells either with PHA and antiCD3/antiCD28 (Figure 5A). On the other hand, upon restimulation, we determined that TemEARLY frequency was significantly high in the CAR19BB T cells activated with PHA (*p* = 0.003, Figure 5A). However, the Tn + Tscm population level decreased in the PHA-activated CAR19BB T cells compared with antiCD3/antiCD28-activated cells (*p* = 0.04, Figure 5A). These data suggest that for the first activation and expansion, there is no significant effect of CAR construct expression for T cell differentiation. Moreover, we determined similar results in CD8+ CAR-T cells (*p* = 0.01, Figure 5B). These data show that PHA stimulation was the pioneer differentiation agent rather than 4-1BB or CD28 costimulatory domains on the CAR constructs.

After characterization of T cell sub-populations upon activation process or CAR construct expression, we wanted to assess exhaustion biomarker expression on the cell populations. We analyzed upregulations of the biomarkers after gating on CD3+ CD4+ or CD8+ EGFRt+ for CAR1928 or CAR199BB-Tcells. TIM3, PD1, and LAG3 expression on CAR-T and untransduced T cell control in the same sample populations were evaluated on day 14 and day 21 with the flow cytometry analysis (Figure 6A). No significant difference was observed in LAG3, TIM3, and PD1 MFI for CD4+ CAR1928 or CAR19BB T cells at day 14 among anti-CD3/anti-CD28 and PHA-activated CD3+ T cells (Figure 6B). No significant difference was observed in LAG3, TIM3, and PD1 MFI of CD8+ CAR1928 or CAR19BB T cells at day 14 among anti-CD3/anti-CD28 and PHA-activated CD3+ T cells (Figure 6C). A similar increase in LAG3, TIM3, and PD1 MFI was observed in the comparison of total CD4+ T cells among anti-CD3/anti-CD28 and PHA-activated CD3+ T cells. Therefore, no statistically significant difference was observed between PHA and anti-CD3/anti-CD28 (Figure 6D). When the LAG3, TIM3, and PD1 MFI values were compared in the comparison of total CD8+ T cells among anti-CD3/anti-CD28 and CD3+ T cells activated with PHA, it was observed that LAG3 value decreased in PHA compared with anti-CD3/anti-CD28. However, there is no big difference (*p =* 0.02, Figure 6E). Second activation and post-culture in CD4+ or CD8+ CAR-T cells with anti-CD3/anti-CD28 or PHA also resulted in statistically insignificant difference of exhaustion biomarkers LAG3, TIM3, and PD1 in CAR1928 and CAR19BB T cells compared with each other on day 7 (day 21 from the first activation) (Figure 6F,G). When anti-CD3/anti-CD28 and PHA reactivations were evaluated in total CD4+ T cells, it was found that TIM3 (*p =* 0.04) and PD1 (*p =* 0.012) rates increased statistically significantly after PHA reactivity (Figure 6H). When anti-CD3/anti-CD28 and PHA reactivations were evaluated in total CD8+ T cells, it was found that LAG3 rates increased statistically significantly after reagents with PHA (*p =* 0.04, Figure 6I). These results show that exhaustion biomarkers increase slightly more especially after reactivation with PHA. Here, we showed that the exhaustion marker levels were not significantly different in either PHA- or aCD3/aCD28-activated T cells encoding CAR1928 or CAR19BB constructs.

### 3.3. Cytotoxic Activity of PHA- or aCD3/aCD28-Activated CAR-T Cells

Next, we aimed to determine the anti-cancer activity of the CAR-T cells encoding CAR1928 or CAR19BB following activation and expansion with PHA or aCD3/aCD28 for 14 days. RAJI B cell lymphoma lines were stained with aCD19 while CAR-T cells in CD3+ T cells were determined with aEGFR. To assess the activation of the cells, upregulation of CD25 and CD107a activation and degranulation markers were determined (Figure 7A). It was observed that CAR19BB a3/28 has high killing efficiency on day 2, especially at 5:1 and 10:1. However, at day 7, the kill rate dropped to almost 0 (*p =* 0.004, Figure 7B). After day 7, it was observed that the rate of cytotoxicity increased to 100% (*p =* 0.0001). For CAR1928 a3/28, almost 100% anti-cancer effect was observed on the 2nd day, while it was observed that it killed 100% at day 7 (*p =* 0.0001, Figure 7B). A very low cytotoxicity rate was observed in CAR19BB PHA, even at a ratio of 10:1 (*ns*). However, even when CAR1928 CAR-T was produced by activating it with PHA, it was observed that it killed close to 100% on the 2nd day and 100% at day 7 (*p =* 0.00001, Figure 7B). Combining PHA-activated CAR1928 CAR-T together with CAR19BB was also observed to effectively kill RAJI (*p =* 0.00001). In the anti-cancer efficacy experiment with CAR-T cells produced by the second activation, RAJI cells became overpopulated after day 7 in all experimental groups (*p =* 0.0001, Figure 7C). The anti-cancer activity was not successful in any CAR-T (PHA or α3/28-activated CAR19BB or CAR1928) cell co-culture experiment (Figure 7C). This shows that CAR-T cells produced by the second activation lost their anti-cancer activity to a large extent. Although it seems to have anti-cancer activity when only CAR1928 is produced by activating T cells with α3/28, RAJI cells continue to proliferate at day 7 (*p =* 0.0001, Figure 7C). Therefore, it was observed that PHA may not suppress the anti-tumor activity with CAR19BB and may show anti-tumor activity effectively with CAR1928.

In the activation of CAR1928 T cells with both anti-CD3/anti-CD28 and PHA (1:1 CAR-T:RAJI), CD25 expression was observed for both CD4+ CAR-T and CD8+ CAR-T. It was observed that CAR19BB exhibited a significant increase in activation by both anti-CD3/anti-CD28 and PHA compared with control CD4+ T and control CD8+ T. As with the anti-CD3/anti-CD28, CD25 and CD107a also increased equally in CAR19BB (*p =* 0.003, Figure 8A and *p =* 0.00001, Figure 8B). In the activation of CAR1928 with both anti-CD3/anti-CD28 and PHA (5:1 CAR-T:RAJI), CD25 expression was observed for both CD4+ CAR-T and CD8+ CAR-T. Compared with CAR1928, it was observed that CAR19BB did not show effective CD25 activation by both anti-CD3/anti-CD28 and PHA. CD107a was also increased at a similarly in CAR19BB compared with CD28. CD8+ CAR-T CAR1928 PHA ratio was higher than the CD8+ T PHA ratio (*p =* 0.022, Figure 8C and *p =* 0.045, Figure 8D). In the activation of CAR1928 with both anti-CD3/anti-CD28 and PHA (10:1 CAR-T:RAJI), CD25 expression was observed to occur successfully for CD4+ CAR-T and CD8+ CAR-T. Compared with CAR1928, it was observed that CAR19BB did not show effective CD25 activation by both anti-CD3/anti-CD28 and PHA. The activation of CD107a was higher in CD8+ CAR-Ts compared with CD4+ CAR-Ts. Although effective CD25 activation was not observed in CAR19BB CAR-T cells, similar activation was observed in CD107a cytotoxic degranulation with anti-CD3/anti-CD28 activated CAR1928 CAR-T cells (*p =* 0.01, Figure 8E and *p =* 0.03, Figure 8F).

## 4. Discussion

The time-dependent efficacy and stability of CAR-T cells, which is the most promising method in some hard to treat cancer treatments, needs to be improved. Currently available CAR-T cell technologies use aCD3/aCD28 microbeads to activate cells. Here we report that PHA can be a good alternative for the activation and expansion of CAR-T cells and remain more stable and transform into memory cell subtypes. 

While the existing cellular immunotherapy treatment methods are ineffective in creating a complete remission, they increase survival. One of the main reasons is that the patient’s immune cells carry low-activity receptors even if they are stimulated against cancer antigens. In this study, we discuss the strategy to produce autologous T lymphocytes for CAR-T, to carry high-affinity receptors, and long term stability. For CAR-T cells to remain long-term and effective in in vitro and in vivo cancer models, it is required to have a central memory feature (Tcm) and stem cell-like memory feature (Tscm). We developed a new alternative CAR-T manufacturing process to replace anti-CD3/anti-CD28 microbeads. We also showed that CAR expression can be increased at the desired rate by transduction after the reactivation. Compared with aCD3/aCD28 microbeads, the proliferation of PHA-activated T cells was significantly increased. Thus, PHA-activated CAR-T cells can multiply more quickly and easily before they can be administered to patients for clinical studies. In both T cell types, the TemEARLY cell population was quite high in cells with PHA activation. More TemEARLY cells means there is a higher proportion of non-exhausted memory cells by PHA stimulation. TemEARLY cells possess a range of effector functions and are predominant in the target tissues. They are more likely to respond to tissue antigen reload than other T cell sub-populations. TemEARLY cells likely perpetuate autoimmune diseases due to their effector functions and relative longevity. Persistent antigen increases the pool of Tem cells, as demonstrated in studies of chronic infections; this would also be true in the context of autoimmune diseases where self-antigens persist [30].

The significance of TemEARLY cells by activation with PHA shows that after CAR-T cell application, CAR-T cells remain in the patient’s blood for a longer time as memory cells and that it is an important defense mechanism in case of relapsing cancer. Apart from the high number of TemEARLY cells, another important result was that the Temra and TemLATE cell populations were less with PHA compared with CD3/CD28 stimulation. Memory cells are of great importance in terms of time-dependent efficacy and stability after CAR-T cell therapy. We also assessed two different CAR constructs compared with each other with different variations to analyze whether aCD3/aCD28 or PHA can make any significant differences depending on diverse CAR constructs. When we compared total CD4+ and CD8+ T cell groups, we also found that memory cells can be preserved after the activation with PHA. It also showed that the cells are significantly farther from the exhausted profile. Our results may also explain why T cells expressing CAR1928 or CAR19BB in anti-cancer studies show higher anti-cancer capacity, especially with a3/28 reactivation.

Combining CAR1928 CAR-T, activated with PHA, together with CAR19BB was also observed to effectively kill the RAJI cells. Therefore, PHA may not suppress the anti-tumor activity very much and may show anti-tumor activity effectively. Although our results show that CAR-T cells produced by the second activation lost their anti-cancer activity to a large extent, it can be mean that reactivation may not be the preferred way of cell killing for PHA-activated CAR-T cells. Thus, we developed a new alternative CAR-T manufacturing process to replace anti-CD3/anti-CD28 microbeads. This is the first result showing that T cells can be transduced by adding CAR lentiviruses in the first activation with PHA compared with anti-CD3/anti-CD28.

CAR-T cell production with PHA had no adverse effects on cell viability compared with aCD3/aCD28. Thus, the use of PHA as an alternative to aCD3/aCD28 in CAR-T cell production can be a parallel line of investigation for future studies. On the other hand, we also noticed that in vivo animal study or anti-cancer capacity of ex vivo CAR-T cells produced from the ALL-patient cohort T cells needed to be more comprehensive. Therefore, we plan in vivo anti-cancer study using the CAR-T cells produced with PHA in ALL animal model. We also showed that CAR expression can be increased at the desired rate by transduction after the reactivation. Based on our initial results, the combined use of CAR19BB and CAR1928 CAR-T cells instead of individual CAR-T cells with the second reactivation may have permanent and long-term anti-cancer efficacy. In future clinical studies, we plan to take advantage of both the CAR19BB-mediated memory T cell ratio and use the high anti-cancer capacity of CAR1928-mediated T cells. In conclusion, we showed that CAR-T cell production with PHA has a similar/better proliferation capacity than CAR-T cell proliferation obtained in anti-CD3/anti-CD28 activation. Following this, we also observed increased Tcm–Tscm ratios and anti-cancer activity of CAR-T cells activated with PHA compared with anti-CD3/anti-CD28-activated CAR-T cells.

## Figures and Tables

**Figure 1 vaccines-11-00313-f001:**
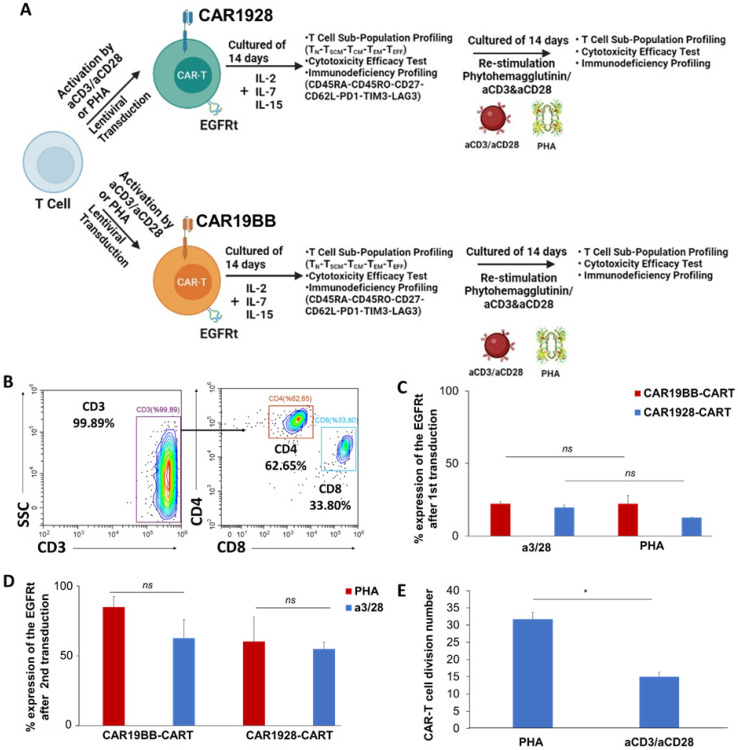
(**A**) The schematic model of the experimental design aims to determine Tcm and Tscm sub-population, proliferation, and cytotoxicity of either CAR1928-T cells (turquoise) or CAR19BB-T cells (red) with CD28 or 4-1BB-based CAR construct upon activation of T cells with Phytohemagglutinin/ aCD3 and aCD28. (**B**) The flow cytometry plots of CD3+ T cell isolation from three different donors. (**C**) The bar graph showing CAR+ T cell frequency with CAR19BB or CAR1928 constructs after activation with PHA or anti-CD3/anti-CD28 by expression of EGFRt. (**D**) The bar graph showing CAR+ T cell frequency with CAR19BB or CAR1928 constructs after second activation and re-transduction with PHA or anti-CD3/anti-CD28 by expression of EGFRt. (**E**) The bar graph shows division numbers of the CAR-expressing total T cells during expansion for 14 days upon stimulation with T cell complete media including either with PHA or aCD3/aCD28. The two-tailed *t*-test statistical significance was represented by * *p* < 0.05 and *ns*: non-significant, and the bars in the graph are the mean +/− standard deviation of the groups.

**Figure 2 vaccines-11-00313-f002:**
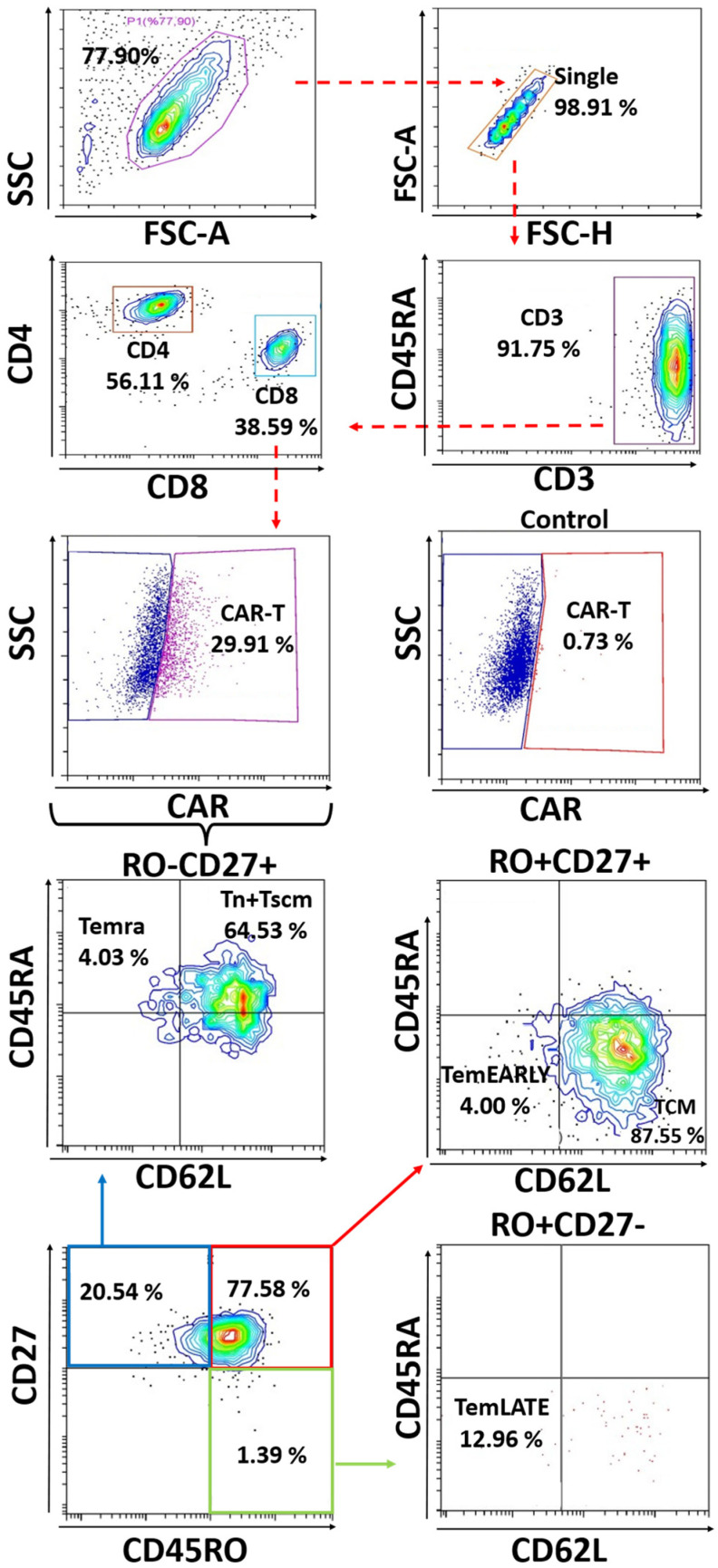
The flow cytometry plots showing a representative set of the T cells sub-population plots from a healthy adult donor in CD4^+^ and CD8^+^ including Tn + Tscm (CD3^+^, CD45RA^+^, CD62L^+^), Tcm (CD3^+^, CD45RA^−^, CD62L^+^), TemEarly (CD3^+^, CD45RO^+^, CD27^+^, CD45RA^−^, CD62L^−^) T cells, TemLate (CD3^+^, CD45RO^+^, CD27^−^, CD45RA^−^, CD62L^−^), and Temra (CD3^+^, CD45RA^+^, CD62L^−^) in healthy donors. The two-tailed *t*-test statistical analysis was performed. Legend: TemEARLY: effector memory early T cells; Tcm: central memory T cells; Tn + Tscm: naïve T cells and stem cell-like memory T cells; Temra: terminally differentiated effector memory T cells; TemLATE: effector memory late T cells.

**Figure 3 vaccines-11-00313-f003:**
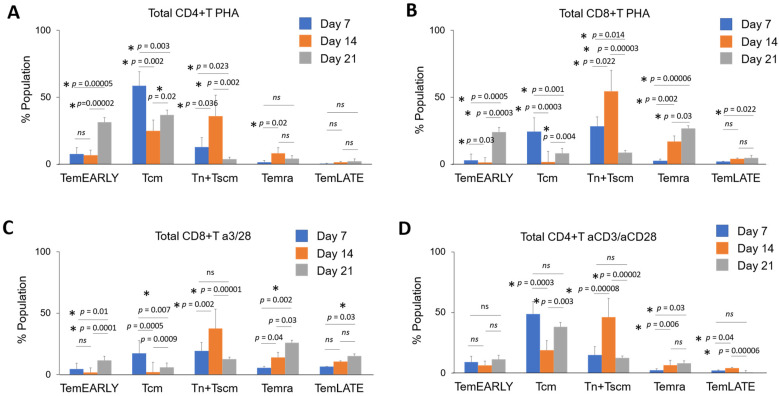
The bar graphs show the quantitated distribution of Tn + Tscm, Tcm, TemEARLY, TemLATE, and Temra sub-populations of total CD4+ or CD8+ T cells in PHA-activated or aCD3/aCD28-activated CD3+ T cells from three healthy donors on day 7, day 14, and day 21. The distribution of the T cell sub-types of (**A**) total CD4+ T cells in PHA-activated CD3+ T cells, (**B**) total CD8+ T cells in PHA-activated CD3+ T cells, (**C**) total CD8+ T cells in aCD3/aCD28-activated CD3+ T cells, and (**D**) total CD4+ T cells in aCD3/aCD28-activated CD3+ T cells. The two-tailed *t*-test statistical significance was represented by * *p* < 0.05 and *ns*: non-significant, and the bars in the graph are the mean +/− standard deviation of the corresponding groups. Legend: TemEARLY: effector memory early T cells; Tcm: central memory T cells; Tn + Tscm: naïve T cells and stem cell-like memory T cells; Temra: terminally differentiated effector memory T cells; TemLATE: effector memory late T cells.

**Figure 4 vaccines-11-00313-f004:**
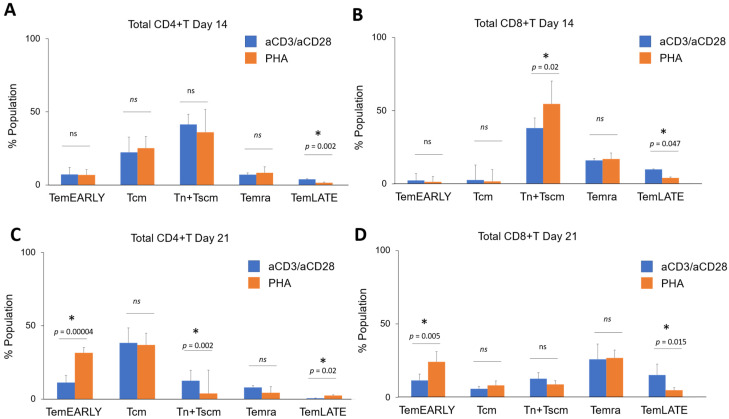
(**A**) The bar graph shows the comparison of Tn + Tscm, Tcm, TemEARLY, TemLATE, and Temra sub-populations of (**A**) total CD4+ T cells and (**B**) total CD8+ T cells either activated with anti-CD3/anti-CD28 or PHA on day 14. The bar graphs show the distribution of T cell sub-populations of (**C**) total CD4+ T cells and (**D**) total CD8+ T cells activated either with anti-CD3/anti-CD28 or PHA on day 21. The two-tailed *t*-test statistical significance was represented by * *p* < 0.05 and *ns*: non-significant, and the bars in the graph are the mean +/− standard deviation of the corresponding groups. Legend: TemEARLY: effector memory early T cells; Tcm: central memory T cells; Tn + Tscm: naïve T cells and stem cell-like memory T cells; Temra: terminally differentiated effector memory T cells; TemLATE: effector memory late T cells.

**Figure 5 vaccines-11-00313-f005:**
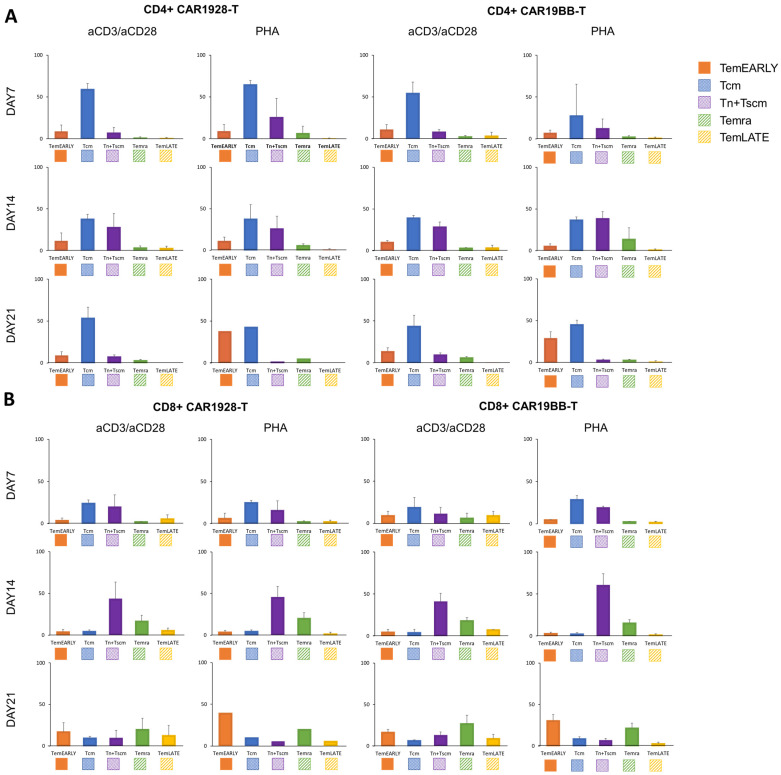
The bar graphs show percentages of T cell sub-populations expressing CAR(CD28) or CAR(4-1BB) in (**A**) the CD4+ or (**B**) the CD8+ T cell population after the first and second activations on days 7, 14, and 21. The two-tailed *t*-test statistical significance, and the bars in the graph are the mean +/− standard deviation of the corresponding groups. Legend: TemEARLY (orange): effector memory early T cells; Tcm (blue): central memory T cells; Tn + Tscm (purple): naïve T cells and stem cell-like memory T cells; Temra (green): terminally differentiated effector memory T cells; TemLATE (yellow): effector memory late T cells.

**Figure 6 vaccines-11-00313-f006:**
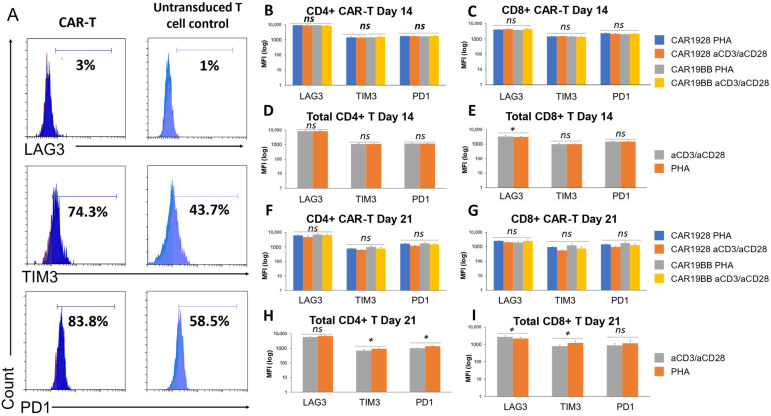
Exhaustion profile of PHA- or aCD3/aCD28-activated CAR-T cells. (**A**) Histograms showing representative plots of LAG3, TIM3, and PD1 expressions of day 7 CAR-T cells and untransduced T cell control in the same sample activated with PHA after gating on CD3+, CD4+, CD8+, and EGFRt+. (**B**) Comparison of MFI values of exhaustion biomarkers (LAG3, TIM3, PD1) of CD4+ CAR1928 or CAR19BB T cells at day 14 in CD3+ T cells activated with anti-CD3/anti-CD28 and PHA. (**C**) Comparison of MFI values at Day 14 of exhaustion biomarkers (LAG3, TIM3, PD1) of CD8+ CAR1928 or CAR19BB T cells in CD3+ T cells activated with anti-CD3/anti-CD28 and PHA. (**D**) Comparison of MFI values on day 14 of exhaustion biomarkers (LAG3, TIM3, PD1) of total CD4+ T cells in CD3+ T cells activated with anti-CD3/anti-CD28 and PHA. (**E**) Comparison of MFI values on day 14 of exhaustion biomarkers (LAG3, TIM3, PD1) of total CD8+ T cells in anti-CD3/anti-CD28 and PHA-activated CD3+ T cells. (**F**) Comparison of MFI values on day 21 of exhaustion biomarkers (LAG3, TIM3, PD1) of total CD4+ CAR-T cells in CD3+ T cells activated with anti-CD3/anti-CD28 and PHA. (**G**) Comparison of exhaustion biomarkers (LAG3, TIM3, PD1) 21st day MFI values of total CD8+ CAR-T cells in CD3+ T cells activated with anti-CD3/anti-CD28 and PHA. (**H**) Comparison of the MFI values of the exhaustion biomarkers (LAG3, TIM3, PD1) of total CD4+ T cells (LAG3, TIM3, PD1) on day 21 among CD3+ T cells activated with anti-CD3/anti-CD28 and PHA. (**I**) Comparison of exhaustion biomarkers of total CD8+ T cells (LAG3, TIM3, PD1) on day 21, among CD3+ T cells activated with anti-CD3/anti-CD28 and PHA. The two-tailed *t*-test statistical significance was represented by * *p* < 0.05 and *ns*: non-significant, and the bars in the graph are the mean +/− standard deviation of the corresponding groups. Legend: MFI: mean fluorescence intensity; LAG3: lymphocyte-activation gene 3; TIM3: T cell immunoglobulin and mucin-domain containing-3; PD1: Programmed cell death protein 1.

**Figure 7 vaccines-11-00313-f007:**
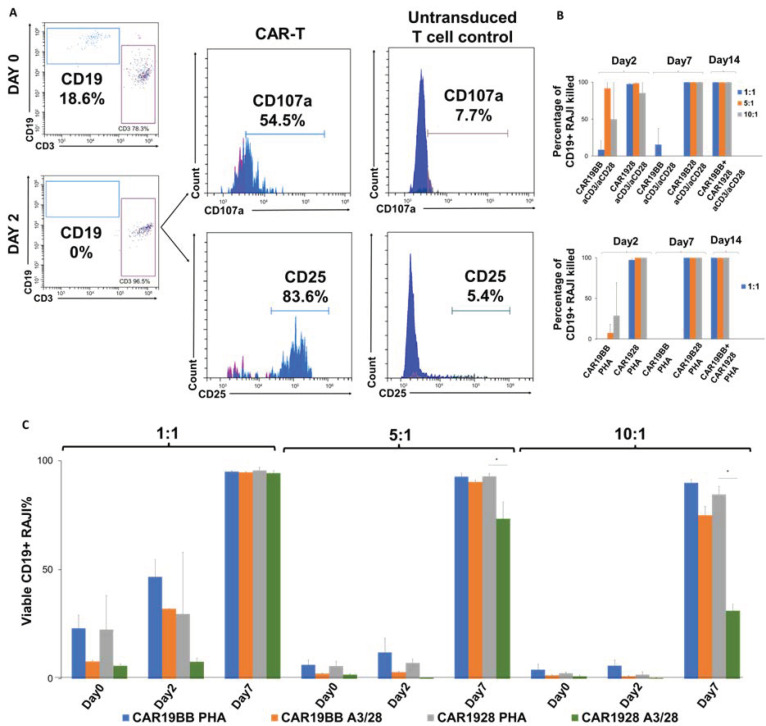
Cytotoxicity capacity of the CAR-T cells. (**A**) Analysis of CD25 activation and CD107a cytotoxic de-granulation biomarkers of CAR-T cells and untransduced T cell control in CD3+ T cells in the sample co-culture by flow cytometry and death of CD19+ RAJI cells after 48 h in CAR-T RAJI co-culture experiments. (**B**) CD19+ RAJI Killing Assay Mortality Rates of CAR19BB and CAR1928 CAR-Ts produced by activating PHA and anti-CD3/anti-CD28 at day 2, 7, and 14. (**C**) Bar graph showing the percentage of living CD19+ RAJI Cells, the anti-cancer activity with CAR-T cells produced by the second activation at day 2 and 7. The two-tailed *t*-test statistical significance was represented by * *p* < 0.05, and the bars in the graph are the mean +/− standard deviation of the corresponding groups. Legend: 1:1, 5:1, or 10:1; Effector CAR-T: Target RAJI cell co-incubated for one-week.

**Figure 8 vaccines-11-00313-f008:**
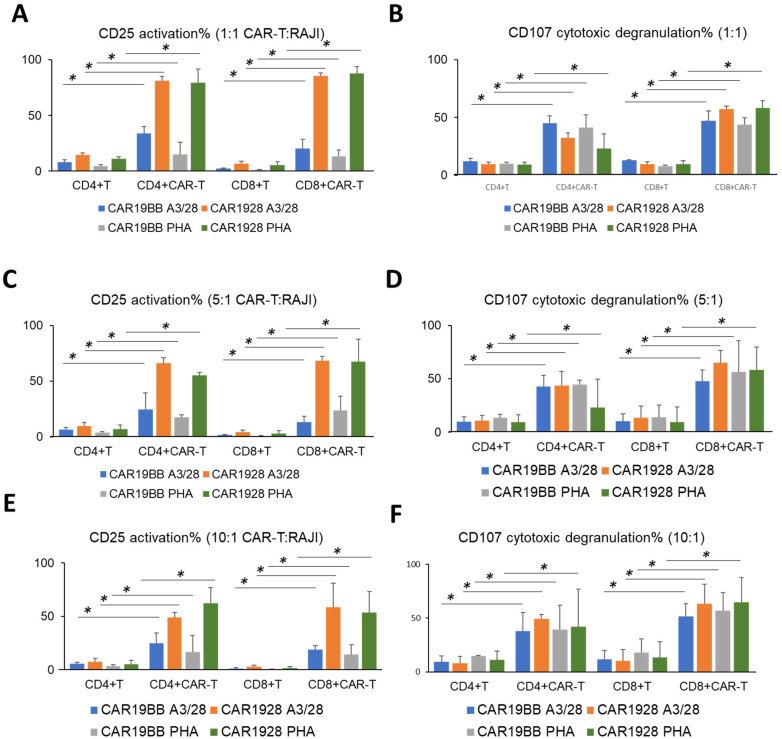
Activation of CAR-T cells with CD25 and CD107a upregulation. Bar graph of (**A**) CD25 activation and (**B**) CD107a cytotoxic degranulation biomarkers of CAR-T cells and control T cells in CD3+ T cells after 48 h in 1:1 CAR-T: RAJI co-culture experiments. Bar graph of (**C**) CD25 activation and (**D**) CD107a cytotoxic de-granulation biomarkers of CAR-T cells and control T cells in CD3+ T cells after 48 h in 5:1 CAR-T: RAJI co-culture experiments. Bar graph of (**E**) CD25 activation and (**F**) CD107a cytotoxic de-granulation biomarkers of CAR-T cells and control T cells in CD3+ T cells after 48 h in 10:1 CAR-T: RAJI co-culture experiment. The two-tailed *t*-test statistical significance was represented by * *p* < 0.05, and the bars in the graph are the mean +/− standard deviation of the groups including CAR19BB aCD3/aCD28 (blue), CAR1928 aCD3/aCD28 (orange), CAR19BB PHA (gray), and CAR1928 PHA (green).

## Data Availability

The data in the paper is unavailable due to privacy.

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
