# Peer review of "CAR-T Cells with Phytohemagglutinin (PHA) Provide Anti-Cancer Capacity with Better Proliferation, Rejuvenated Effector Memory, and Reduced Exhausted T Cell Frequencies"

_vaccines, 2023, doi:10.3390/vaccines11020313_

Round 1
Reviewer 1 Report
In the present study, Gulden et.al. use PHA as an activation signal to boost the functional properties of CAR-T cells for anticancer therapy. The study is well-planned and well-executed, and the manuscript is generally well-written with adequate references.
I only have two major concerns:
The first is the lack of in vivo validation of the efficacy of the method. Although I like the extensive validations of the functional properties of the T cell subtypes, I expected to see some experiments in a cohort of human patients. Will it be possible for the authors to do so?
The second issue I have with the manuscript is the extensive use of pie charts for data display. Pie charts are generally accepted as a pretty bad form of data display, particularly if the number of variables is more than 2 or 3. I would strongly suggest the authors remove all the pie charts from the manuscript and replace them with bar plots or heatmaps.
Specific comments:
Figure 2: Please mention which one is 4-1BB or CD28 - based CAR construct in the legend.
Figures 7, 8, 9, and 10: Please remove the pie charts. The bar plots convey the message well enough.
Figure 14: Please replace the pie charts with bar plots or heatmaps.
Author Response
Rebuttal:
ANSWERS TO THE COMMENTS:
We thank the Reviewers for this revision of our manuscript and the positive comments that the manuscript is now much improved. We have made additional revisions for clarity and conciseness throughout the text. Here, you can find the answers given to the comments of the referees. We have tried to answer the number of points that the referees have outlined. These points are discussed below following the reviewer’s comments. We hope that the points amended in the paper comprehensively would be satisfactory enough to correspond to the comments of the referees. We addressed the remaining questions of the Reviewers in point-by-point responses below and opened “track each change” to show the revisions and corrections on the manuscript.
Reviewer 1’s comments:
Comment: In the present study, Gulden et.al. use PHA as an activation signal to boost the functional properties of CAR-T cells for anticancer therapy. The study is well-planned and well-executed, and the manuscript is generally well-written with adequate references. I only have two major concerns: The first is the lack of in vivo validation of the efficacy of the method. Although I like the extensive validations of the functional properties of the T cell subtypes, I expected to see some experiments in a cohort of human patients. Will it be possible for the authors to do so?
Answer: We thank the reviewer for this comment. We also noticed that in vivo animal study or anti-cancer capacity of ex vivo CAR-T cells produced from the ALL-patient cohort T cells needed to be more comprehensive. Therefore, we plan in vivo anti-cancer study using the CAR-T cells produced with PHA in ALL animal models. We have mentioned this comment and the plan in the last paragraph of the discussion section.
Comment: The second issue I have with the manuscript is the extensive use of pie charts for data display. Pie charts are generally accepted as a pretty bad form of data display, particularly if the number of variables is more than 2 or 3. I would strongly suggest the authors remove all the pie charts from the manuscript and replace them with bar plots or heatmaps.
Answer: Thanks very much for this critical comment from the reviewer. We have removed all the pie charts in Figure 14 in the submitted manuscript and replaced them with the bar plots in Figure 5 in the revised manuscript. Furthermore, we have combined the first five figures into Figure 1 in the revised manuscript. Moreover, we combined Figures 11-13 into Figure 4 in the revised manuscript.
Comment: Specific comments:
- Figure 2: Please mention which one is 4-1BB or CD28 - based CAR construct in the legend.
- Figures 7, 8, 9, and 10: Please remove the pie charts. The bar plots convey the message well enough.
- Figure 14: Please replace the pie charts with bar plots or heatmaps.
Answer: Thanks for these specific comments.
- Figure 2 in the submitted manuscript is combined into Figure 1 in this revised manuscript. The legend is revised as “Figure 1. A. The experimental setup aims to determine Tcm and Tscm sub-population, proliferation, and cytotoxicity of either CAR1928-T cells (turquoise) or CAR19BB-T cells (red) with CD28 or 4-1BB-based CAR construct upon activation of T cells with Phytohemagglutinin/ aCD3&aCD28.”
- We removed the pie charts in Figures 7-10 in the submitted manuscript and combined the bar graphs in these figures into Figure 3 in the revised manuscript.
- We have removed all the pie charts in Figure 14 in the submitted manuscript and replaced them with the bar plots in Figure 5 in the revised manuscript.
Reviewer 2’s comments:
Comment: The authors used well-established and certainly not novel peripheral blood cells activation methods, such as activation with PHA and IL15. On the other hand, it is appreciable that this activation strategy has been applied to CAR-Ts to solve and possibly improve the therapeutic approach in the various hematologic cancer diseases listed comprehensively. The authors show that they have deep knowledge of the basal methods, which they have exploited well by applying them to innovate. The introduction is well written, influent language, and very well explained with good references. The materials and methods are also complete, clear, and follow the journal's guidelines. Statistical analysis is robust and is performed with an appropriate test for the data, and SPSS is good software. In particular, the isolation and immunoprofiling sub-population gating strategy in cytometry is well described, and the experimental setup is clear. Good description of overcoming critical issues. The strategy of increasing the MOI ratio to 5-10 to provide more than 20% expression in the cells in the second transduction process, when the authors activated with PHA, ensured the production of CAR-T cells effectively. Also in the cytotoxicity study, the strategy of combining CAR1928 CAR-T with CAR19BB CAR-T to achieve both extended cells killing capacity for up to 7 days on RAJI. The authors point out that PHA does not show the same anti-cancer capacity in both constructs. Another relevant point is that PHA gives no adverse effects when compared with aCD3/aCD28. In light of the results, I agree with the authors that PHA may be a viable alternative to aCD3/aCD28 if confirmed by in vivo studies.
Answer: We thank Reviewer 2 for the positive comments about the manuscript. We believe that the manuscript will enable researchers to focus on how CAR-T cell production capacity can be improved with preserved or improved anti-cancer activity along with the memory phenotype. We also noticed that in vivo animal study with the PHA-induced CAR-T cells needed to be more comprehensive and we plan in vivo anti-cancer study using the CAR-T cells in an ALL-animal model.
Reviewer 3’s comments:
Comment: In this paper, different memory t cell population ratios and cytotoxic activity were assessed in the PHA or aCD3/aCD28-activated CAR-T cells with two different CAR(aCD19 scFv- 104 CD8-(CD28 and/or 41BB)-CD3z-EGFRt) constructs. The paper would need English language revision. There are several mistakes and descriptions are often not clear and incomplete.
Answer: We thank Reviewer 3 for the comment. We have revised the manuscript about this issue. We indicated the mistakes and descriptions along with the English of the manuscript and opened “track changes” in Word to show all revisions.
Comment: All figure legends need detailed descriptions. There are some missing details which would need to be added, including a detailed description of statistical analysis, tests, and results.
Answer: Thanks for these specific comments. We have added all details including acronyms, statistic information, and test numbers into all figure legends.
Comment: Figure 2 would rather be used as a schematic model of the experimental design and therefore not useful to represent results. Moreover, the legend needs to indicate all the acronyms, if not indicated before. Indeed authors indicate two distinct subpopulation to be studied while, for instance, in this figure and the results shown in figure 6, there are different acronyms, but are not specified and it is not clear to which population are refereed (Tn-Tem-Tef).
Answer: Thank you the Reviewer for this advice as we have merged Figures 1-5 in the submitted manuscript into Figure 1 of the revised manuscript. The previous Figure 2 is positioned to be Figure 1A showing the schematic model of the experimental setup of the study and revised in the legend.
Comment: Figures 3 and 4 show the same phenomenon, at different time points and would rather be grouped in one figure, comparing the different time points and using the same graph type and labels, which are now inverted. Moreover, the two-time points need to be compared and statistically analyzed.
Answer: In Figure 3 (now Figure 1C of the revised manuscript), we tested the lentiviral transduction capability of PHA-induced T cells with CAR1928 or CAR19BB expression. We showed that PHA-induced T cells can be transduced with CAR-encoding lentiviruses without statistical difference regarding CAR expression on the aCD3/aCD28-induced T cells. In Figure 4 (now Figure 1D of the revised manuscript), we asked whether we can increase CAR expression using 2nd lentiviral transduction after reactivation of the T cells with the same activation reagents including aCD3/aCD28 and PHA. We showed that both T cell restimulation approaches could achieve upregulation of the CAR expression with 2nd transduction. This suggests that there is no difference in the production of CAR-T cells with lentiviral transduction after the second stimulation. We mentioned this finding in the result section. Therefore, we did not compare Figures 1C and 1D as the questions are different.
Comment: Figure 5: figure legend is not sufficiently detailed and needs to be improved with further statistical description, type of test used, and samples used. Moreover, I would suggest merging figures 3, 4, and 5 in one figure with more panels showing similar and complementary results.
Answer: As Reviewer advised, we have detailed the legend of figure 5 (now Figure 1E of the revised manuscript). Furthermore, we have combined Figures 1-5 of the submitted manuscript into Figure 1 in the revised manuscript. We also removed the pie charts in Figures 7-10 of the submitted manuscript and merged the figures into Figure 3 of the revised manuscript. Moreover, we combined Figures 11-13 into Figure 4 in the revised manuscript.
Comment: Figure 6: there are several populations indicated and studied, which were not named before. The acronyms are reported in a very floppy way, and this is very confusing. The authors should consistently refer to the different populations studied. Even if the plots are representative of independent experiments, the statistical analysis should be mentioned and clearly stated.
Answer: Thanks very much for this critical comment from the reviewer. The acronyms were revised and detailed in Figure 2 of the revised manuscript. Figure 2 is representative of an experiment; however, we added statistical analysis details of all bar graphs analyzed from Figure 2 into Figure 3 and Figure 4 of the revised manuscript.
Comment: In figure 7, the scale bars are missing. Does the percentage refer to the percentage of positive cells by flow cytometry? While figure 6 shows the percentage and gating strategy of cells from healthy donors, it is not clear which specimens are used afterward. There are several (too many) figures following this one, showing many results, all missing proper detailed statistics, description of the population, scale bars, and comparisons between the different populations. Pie charts do not add anything to the bar plots already showing the data and perhaps should be moved to a supplementary section.
Answer: We have removed all the pie charts in Figure 14 in the submitted manuscript and replaced them with the bar plots and scale bars in Figure 5 in the revised manuscript.
Comment: In general, the paper would need major revision, trying to show and describe the results, use fewer figures, and perhaps use a supplementary results section to report all these results.
Answer: We have appreciated these critical comments of Reviewer 2. We have made additional revisions for clarity and conciseness throughout the text. We used fewer figures and detailed legends for easy understanding.

Reviewer 2 Report
The authors used well-established and certainly not novel peripheral blood cells activation methods, such as activation with PHA and IL15. On the other hand, it is appreciable that this activation strategy has been applied to CAR-Ts to solve and possibly improve the therapeutic approach in the various hematologic cancer diseases listed comprehensively.
The authors show that they have deep knowledge of the basal methods, which they have exploited well by applying them to innovate
The introduction is well written, in fluent language, and very well explained with good references.
The materials and methods are also complete, clear, and follow the journal's guidelines.
Statistical analysis is robust and is performed with an appropriate test for the data, and SPSS is good software.
Results
In particular, the isolation and immunoprofiling sub-population gating strategy in cytometry is well described, and the experimental setup is clear.
Good description of overcoming critical issues. The strategy of increasing the MOI ratio to 5-10 to provide more than 20% expression in the cells in the second transduction process, when the authors activated with PHA, ensured the production of CA-T cells effectively.
Also in the cytotoxicity study, the strategy of combining CAR1928 CAR-T with CAR19BB CAR-T to achieve both extended cell killing capacity for up to 7 days on RAjii. The authors point out that PHA does not show the same anti-cancer capacity in both constructs.
Another relevant point is that PHA gives no adverse effects when compared with aCD3/aCD28.
In light of the results, I agree with the authors that PHA may be a viable alternative to aCD3/aCD28 if confirmed by in vivo studies.
Author Response

(The authors gave the same response as above.)

Reviewer 3 Report
In this paper, different memory t cell population ratios and cytotoxic activity were assessed in the PHA or aCD3/aCD28-activated CAR-T cells with two different CAR(aCD19 scFv- 104 CD8-(CD28 and/or 41BB)-CD3z-EGFRt) constructs. The
The paper would need English language revision. There are several mistakes and descriptions are often not clear and incomplete.
All figure legends need detailed description. There are some missing details which would need to be added, including detailed description of statistical analysis, tests and results.
Figure 2 would rather be used as a schematic model of the experimental design and therefore not useful to represent results. Moreover, the legend needs to indicate all the acronyms, if not indicated before. Indeed authors indicate two distinct subpopulation to be studied while, for instance, in this figure and the results shown in figure 6, there are different acronyms, but are not specified and it is not clear to which population are refereed to (Tn-Tem-Tef).
Figure 3 and 4 show the same phenomenon, at different time points and would rather be grouped in one figure, comparing the different time points and using the same graph type and labels, which are now inverted. Moreover, the two time points need to be compared and statistically analyzed.
Figure 5: figure legend is not sufficiently detailed and needs to be improved with further statistical description, type of test used, and samples used. Moreover, I would suggest to merge figure 3, 4 and 5 in one figure with more panels showing similar and complementary results.
Figure 6: there are several populations indicated and studied, which were not named before. The acronyms are reported in a very floppy way, and this is very confusing. The authors should consistently refer to the different population studied. Even if the plots are representative of independent experiments, the statistical analysis should be mentioned and clearly stated.
In figure 7, the scale bars are missing. Does the percentage refer to percentage of positive cells by flow cytometry? While figure 6 shows the percentage and gating strategy of cells from healthy donors, it is not clear which specimens are used afterwards. There are several (too many) figures following this one, showing many results, all missing proper detailed statistics, description of the population, scale bars, comparisons between the different population. Pie chart do not add anything to the bar plots already showing the data and perhaps should be moved to a supplementary section.
In general, the paper would need major revision, trying to more clearly show and describe the results, use less figures, and perhaps use a supplementary results section to report all these results.
Author Response

(The authors gave the same response as above.)

Round 2
Reviewer 3 Report
Lines 213-214-215 please move to methods
Line 223 please indicate mean +/- standard deviation, please indicate statistical test used
Figure 1B please report one donor as representative plot of the three donors analysed, and restore the scale bars on the flow cytometry plots (not visible).
Figure 1C and D: please leave the x and y axis, please adjust the scale of the two plots using the same scale and indicate the statistical test used and mean +/- standard deviation for both graphs in the figure legend. If C and D indicate the % of expression of the EGFR expression, then indicate it in the legend next to the y axis.
Figure 1E please indicate the exact p value and the type of test used. It is not clear what the value on the y axis indicates: cell division percentage?
Figure 2 shows complete scale bars with numbers indicating the logarithmic expression of the analysed markers. Although probably the authors are showing a representative set of plots, statistics tests needs to be indicated.
Figure 3, 4, 5: please indicate what the numbers represents: mean+/- standard deviation should be indicated for the different experiments performed. Please indicate type of statistical test used in the figure legend and exact p values. Please re-insert y and x axis. Figure 5 is very small: while the y axis legend is huge it is very difficult to visualize the different population names.
Line 320 change “statistically” with “significantly” increased, and indicate statistic test used and p value
Line 347 or 356: please use consistent acronym indication for 4-1BB.
Figure 6: A) the gating strategy for the histograms is not clear, please adjust; what is “control T “ exactly? Please specify. From B to I: on the x axis, the name of the marker is indicated and then the MFI reported on the y axis. This is not clear, please adjust.
Figure 7A: please re-insert scale bars on the flow cytometry plots and correct Kontrol with control. Please indicate what the control T cells are. 7B : please indicate legend. Please indicate means +/- standard deviation.
Figure 8: the bars for the statistics on top of the graph bars are including all the bars, it is not clear which comparison the significance star is for. Please be clearer with the statistics used and the comparisons done. Please insert the labels on the y axis. The sentence at line 469-470 is not clear. Please specify the comparisons and relative p values.
Author Response
We thank the Reviewer for this revision of our manuscript and the positive comments that the manuscript is now much improved. We have made additional revisions for clarity and conciseness throughout the text. Here, you can find the answers given to the comments of the referees. We have tried to answer the number of points that the referees have outlined. These points are discussed below following the reviewer’s comments. We hope that the points amended in the paper comprehensively would be satisfactory enough to correspond to the comments of the referees. We addressed the remaining questions of the Reviewers in point-by-point responses below and opened “track each change” to show the revisions and corrections on the manuscript.
